# Contrastive learning of global and local features for medical image segmentation with limited annotations

**Krishna Chaitanya**    **Ertunc Erdil**    **Neerav Karani**    **Ender Konukoglu**

Computer Vision Lab, ETH Zurich
Sternwartstrasse 7, Zurich 8092, Switzerland

`krishna.chaitanya@vision.ee.ethz.ch`

## Abstract

A key requirement for the success of supervised deep learning is a large labeled dataset - a condition that is difficult to meet in medical image analysis. Self-supervised learning (SSL) can help in this regard by providing a strategy to pre-train a neural network with unlabeled data, followed by fine-tuning for a downstream task with limited annotations. Contrastive learning, a particular variant of SSL, is a powerful technique for learning image-level representations. In this work, we propose strategies for extending the contrastive learning framework for segmentation of volumetric medical images in the semi-supervised setting with limited annotations, by leveraging domain-specific and problem-specific cues. Specifically, we propose (1) novel contrasting strategies that leverage structural similarity across volumetric medical images (domain-specific cue) and (2) a local version of the contrastive loss to learn distinctive representations of local regions that are useful for per-pixel segmentation (problem-specific cue). We carry out an extensive evaluation on three Magnetic Resonance Imaging (MRI) datasets. In the limited annotation setting, the proposed method yields substantial improvements compared to other self-supervision and semi-supervised learning techniques. When combined with a simple data augmentation technique, the proposed method reaches within 8% of benchmark performance using only two labeled MRI volumes for training, corresponding to only 4% (for ACDC) of the training data used to train the benchmark. The code is made public at https://github.com/krishnabits001/domain_specific_cl.

## 1 Introduction

Supervised deep learning provides state-of-the-art medical image segmentation [47, 39, 31, 32], *when large labeled datasets are available*. However, assembling such large annotated datasets is challenging, thus methods that can alleviate this requirement are highly desirable. Self-supervised learning (SSL) is a promising direction to this end: it provides a pre-training strategy that relies only on unlabeled data to obtain a suitable initialization for training downstream tasks with limited annotations. In recent years, SSL methods [16, 44, 42, 21] have been highly successful for downstream analysis of not only natural images [48, 35, 20], but also medical images [71, 5, 60, 30, 52, 11].

In this work, we focus on contrastive learning [12, 40, 4, 27, 57, 43], a successful variant of SSL. The intuition of this approach is that different transformations of an image should have similar representations and that these representations should be dissimilar from those of a different image. In practice, a suitable *contrastive loss* [24, 12] is formulated to express this intuition and a neural network (NN) is trained with unlabeled data to minimize this loss. The resulting NN extracts image representations that are useful for downstream tasks, such as classification or object detection, and

constitutes a good initialization that can be fine-tuned into an accurate model, even with limited labeled examples.

Despite its success, we believe that two important aspects have been largely unexplored in the existing contrastive learning literature that can improve the current state-of-the-art with respect to medical image segmentation. Firstly, most works focus on extracting global representations and do not explicitly learn distinctive local representations, which we believe will be useful for per-pixel prediction tasks such as image segmentation. Secondly, the contrasting strategy is often devised based on transformations used in data augmentation, and do not necessarily utilize a notion of similarity that may be present across different images in a dataset. We believe that a domain-specific contrasting strategy that leverages such inherent structure in the data may lead to additional gains by providing the network with more complex similarity cues than what augmentation can offer.

In this work, we aim to fill these gaps in the contrastive learning literature in the context of segmentation of volumetric medical images and make the following contributions. 1) We propose new domain-specific contrasting strategies for volumetric medical images, such as Magnetic Resonance Imaging (MRI) and Computed Tomography (CT). 2) We propose a local version of contrastive loss, which encourages representations of local regions in an image to be similar under different transformations, and dissimilar to those of other local regions in the same image. 3) We evaluate the proposed strategies on three MRI datasets and find that combining the proposed global and local strategies consistently leads to substantial performance improvements compared to no pre-training, pre-training with pretext tasks, pre-training with global contrastive loss, as demonstrated to yield state-of-the-art accuracy in [12], as well as semi-supervised learning methods. 4) We investigate if pre-training with the proposed strategies has complementary benefits to other methods for learning with limited annotations, such as data augmentation and semi-supervised training.

## 2 Related works

Recent works have shown that SSL [16, 44, 42, 21] can learn useful representations from unlabeled data by minimizing an appropriate unsupervised loss during training. Resulting network constitutes a good initialization for downstream tasks. For brevity, we discuss only the SSL literature relevant to our work. This can be coarsely classified into two categories.

**Pretext task-based** methods employ a task whose labels can be freely acquired from unlabeled images, to learn useful representations. Examples of such pretext tasks include predicting image orientation [21], inpainting [44], context restoration [11], among many others [18, 16, 42, 61, 62, 17].

**Contrastive learning** methods employ a contrastive loss [24] to enforce representations to be similar for similar pairs and dissimilar for dissimilar pairs [57, 25, 40, 12, 54]. Similarity is defined in an unsupervised way, mostly through using different transformations of an image as similar examples, as was proposed in [18]. [43, 26, 27, 4] maximize mutual information (MI), which is very similar in implementation to contrastive loss, as pointed in [55]. In [27, 4, 26], MI between global and local features from one or more levels of an encoder network is maximized. Authors in [56] used domain-specific knowledge of videos while modeling the global contrastive loss. Others leverage memory bank [57] or momentum contrast [25, 40] to use more negative samples per batch.

The proposed work differs from existing contrastive learning approaches in multiple ways. Firstly, previous works focus on encoder type architectures, used in image-wise prediction tasks, while we focus on encoder-decoder architectures used for pixel-wise predictions. Secondly, works that consider local representations learn global and local level representations simultaneously, where the aim is to facilitate learning of better image-wide representations. In our work, we aim to learn local representations to complement image-wide representations, providing the ability to distinguish different areas in an image to a decoder. As such the minimization objective is different. Thirdly, while learning global representations, we integrate domain knowledge from medical imaging in defining the set of similar image pairs that goes beyond different transformations of the same image.

Other relevant directions that leverage unlabeled data to address the limited annotation problem are semi-supervised learning and data augmentation. **Semi-supervised learning** methods make use of unlabeled along labeled data [10, 66, 23, 65, 37, 33, 46, 36, 49, 53, 41] and successful approaches for medical image analyses include (i) self-training [58, 38, 6] and (ii) adversarial training [63]. Extensive **data augmentation** has also been shown to be beneficial for medical image analyses in limited label

settings. Augmentation through random affine [14], elastic [47] and contrast transformations [29, 45], Mixup [59, 19] and synthetic data generation via GANs [22, 15, 50, 8] have been explored. Recent works also leveraged unlabeled data for augmentation either through image registration [64] or, in more general way, optimizing the augmentation procedures for a given task [9].

## 3 Methods

Our investigation is based on the contrastive loss shown to achieve state-of-the-art performance in [12]. We present first this loss briefly as "global contrastive loss" before our contributions.

### 3.1 Global contrastive loss

For a given encoder network $e(\cdot)$, the contrastive loss is defined as:

$$l(\tilde{x}, \hat{x}) = -\log \frac{e^{\text{sim}(\tilde{z}, \hat{z})/\tau}}{e^{\text{sim}(\tilde{z}, \hat{z})/\tau} + \sum_{\bar{x} \in \Lambda^-} e^{\text{sim}(\tilde{z}, g_1(e(\bar{x})))/\tau}}, \ \tilde{z} = g_1(e(\tilde{x})), \ \hat{z} = g_1(e(\hat{x})). \quad (1)$$

Here, $\tilde{x}$ and $\hat{x}$ are two differently transformed versions of the same image $x$, i.e. $\tilde{x} = \tilde{t}(x)$ and $\hat{x} = \hat{t}(x)$ where $\tilde{t}, \hat{t} \in \mathcal{T}$ are simple transformations as used in [24, 12, 4, 18, 54], such as crop followed by color transformations, and $\mathcal{T}$ is the set of such transformations. These two images are treated as similar and their representations are encouraged to be similar. In contrast, the set $\Lambda^-$ consists of images that are dissimilar to $x$ and its transformed versions. This set may include all images other than $x$, including their possible transformations. Minimizing the loss $l(\tilde{x}, \hat{x})$ increases the similarity between the representations of $\tilde{x}$ and $\hat{x}$, denoted by $\tilde{z}$ and $\hat{z}$, while increasing the dissimilarity between the representation of $x$ and those of dissimilar images. Note that the representations used in the loss are extracted after appending $e(\cdot)$ with $g_1(\cdot)$, a shallow fully-connected network with limited capacity, also referred to as "projection head" in [12]. The presence of $g_1(\cdot)$ allows $e()$ some flexibility to also retain information regarding the transformations, as was empirically shown in [12, 13]. Lastly, similarity in the representation space is defined via the cosine similarity between two vectors, i.e. $\text{sim}(a, b) = a^T b / \|a\| \|b\|$, and $\tau$ is a temperature scaling parameter. Equation 1 only defines the loss for a given pair of similar images. Using this loss, the global contrastive loss is defined as:

$$L_g = \frac{1}{|\Lambda^+|} \sum_{\forall (\tilde{x}, \hat{x}) \in \Lambda^+} [l(\tilde{x}, \hat{x}) + l(\hat{x}, \tilde{x})], \quad (2)$$

where $\Lambda^+$ is the set of all similar pairs of images that can be constructed from a given set of images $\mathbf{X}$. Authors in [12] construct similar pairs by randomly sampling $\tilde{t}$ and $\hat{t}$ from $\mathcal{T}$ and applying them to any given image $x \in \mathbf{X}$. When the global contrastive loss is optimized using mini-batches, each batch is composed of a number of similar image pairs. While computing $l(\tilde{x}, \hat{x})$ for each pair, images in the other pairs form $\Lambda^-$.

The specific definitions of the sets of similar image pairs $\Lambda^+$ and dissimilar images $\Lambda^-$ are the guiding components in the global contrastive loss. Definitions of these sets can be done in a number of ways. Integrating domain and problem-specific information in these definitions have the potential to improve the effectiveness of the resulting self-supervised learning process and impact performance gains on downstream tasks [28]. With this motivation, we investigate different definitions of these sets, focusing on image segmentation for medical applications. Particularly, we leverage domain-specific knowledge for providing global cues for learning with volumetric medical images and problem-specific knowledge for providing local cues for image segmentation. Below, we describe the notions of being similar and dissimilar that we investigated and the associated definitions of sets $\Lambda^+$ and $\Lambda^-$. In the following discussion, we focus on 2D encoder-decoder architectures that are used for image segmentation, such as the UNet architecture [47]. In such architectures, we use the encoder to extract the global representation and the decoder layers to extract the complementary local representation.

### 3.2 Leveraging structure within medical volumes for global contrastive loss

A distinctive aspect of medical imaging, in particular of MRI and CT, is that volumetric images of the same anatomical region for different subjects have similar content. Moreover, such images, especially when they are acquired with the same modality and capture the same field-of-view, can be

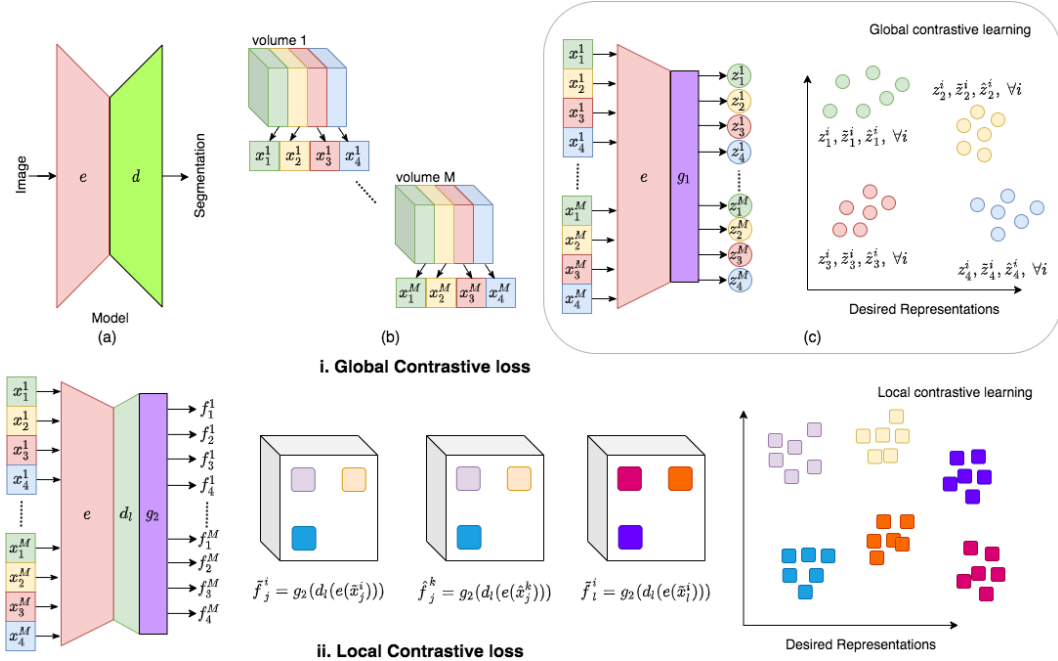

Figure 1: (i) Sketch of global contrastive loss used for pre-training the encoder $e$ with dense layers $g_1$. Here, the number of partitions $S$ per volume are 4. $x_s^i$ denotes the image from partition $s$ of volume $i$, and $z_s^i$ the corresponding global representation. (ii) Sketch of local contrastive loss used for pre-training the decoder $d_l$ with $1 \times 1$ convolutional layers $g_2$, with frozen weights of encoder $e$ obtained from the previous training stage. $f_s^i$ is the corresponding feature map for the image $x_s^i$.

roughly aligned relatively easily using linear registration, for instance as implemented in Elastix [34] Sometimes, this can even be done without using an external registration algorithm, by directly using the transformation matrices from the header files of medical images. Post-alignment, corresponding 2D slices in different volumes capture similar anatomical areas and as such, information in such slices can be considered to be similar. For all experiments in this work, the volumetric images provided in the challenge datasets were already roughly aligned, so we did not perform any registration step.

We leverage the similarity across corresponding slices in different volumes as an additional cue for defining $\Lambda^+$ and $\Lambda^-$ in the global contrastive loss. In the rest of the paper, we refer to a volumetric image by a volume and to a 2D slice by an image, unless otherwise stated. Suppose we have $M$ volumes, each composed of $Q$ images and roughly aligned rigidly. We group the $Q$ images of each volume into $S$ partitions, each containing consecutive images, and denote an image from the $s^{\text{th}}$ partition of the $i^{\text{th}}$ volume by $x_s^i$ (see Fig. 1.i.b). We use the same grouping in all volumes and based on the assumed alignment, corresponding partitions in different volumes can be considered to capture similar anatomical areas. Accordingly, our main hypothesis is that $x_s^i$ and $x_s^j$ can be considered to be similar while training for contrastive learning. Similar to Sec. 3.1, we denote different transformations of an image $x_s^i$ with $\tilde{x}_s^i = \tilde{t}(x_s^i)$ and $\hat{x}_s^i = \hat{t}(x_s^i)$. Representations are extracted with an encoder $e(\cdot)$ followed by $g_1(\cdot)$, and denoted with $z$, e.g. $\tilde{z}_s^i = g_1(e(\tilde{x}_s^i))$. In our explanations, we focus on mini-batch optimization and describe how each mini-batch is constructed.

**Random strategy** ($G^R$): We refer to the application of the original idea (Sec. 3.1) as the "random strategy". As in [12, 25], we form a batch by randomly sampling $N$ images across all volumes and applying a pair of random transformations, $\tilde{t}$ and $\hat{t}$, to each image. $\Lambda^+$ is composed of $(\tilde{x}_s^i, \hat{x}_s^i)$ pairs and for each related pair, the unrelated image set $\Lambda^-$ consists of all the remaining $2N - 2$ images.

**Proposed strategies**: We build upon $G^R$ with two contrasting strategies that leverage similarity of corresponding partitions in different volumes. For both these strategies, we form batches by first randomly sampling $m < M$ volumes. Then, for each sampled volume, we sample one image per partition resulting in $S$ images per volume. Next, we apply a pair of random transformations on each sampled image $(x_s^i)$ to get the transformed images $(\tilde{x}_s^i, \hat{x}_s^i)$ and add them to the batch.

**Strategy** $G^{D-}$: Here, we restrict contrasting images from corresponding partitions in different volumes as such images likely contain similar global information and treating them as dissimilar may be detrimental. For a given related pair $(\tilde{x}_s^i, \hat{x}_s^i)$ from partition $s$, we enforce $\Lambda^-$ to contain images *only from partitions other than* $s$: $\Lambda^- = \{x_k^l, \tilde{x}_k^l, \hat{x}_k^l \,|\forall k \neq s, \forall l\}$. The set of similar pairs, $\Lambda^+$, contains three pairs: $(x_s^i, \tilde{x}_s^i), (x_s^i, \hat{x}_s^i)$ and $(\tilde{x}_s^i, \hat{x}_s^i)$, for each partition and sampled volume.

**Strategy** $G^D$: Here, we further incentivize similar representations for images coming from corresponding partitions of different volumes, thus encouraging partition-wise representation clusters (as illustrated in Fig. 1.i.c). Compared to $G^R$, $G^D$ complements random transformations with real images from other volumes for which representations should be similar. Accordingly, $\Lambda^+$ contains pairs of images from corresponding partitions across volumes $(x_s^i, x_s^j)$, as well as transformed versions $(\tilde{x}_s^i, \hat{x}_s^j)$, in addition to the pairs described in $G^{D-}$. For a given pair, $\Lambda^-$ remains the same as in $G^{D-}$.

## 3.3 Local contrastive loss

The global contrastive loss incentivizes *image-level* representations that are similar for similar images and dissimilar for dissimilar images. This strategy is useful for downstream tasks such as object classification, that require distinctive image-level representations. Downstream tasks that involve pixel-level predictions, for instance image segmentation, may additionally require distinctive local representations to distinguish neighbouring regions. A suitable local strategy, in combination with the global strategy, may be more apt for such downstream tasks.

With this motivation, we propose a self-supervised learning strategy that encourages the decoder of an encoder-decoder network (Fig. 1.ii) to extract distinctive local representations to complement global representations extracted by the encoder. Specifically, we train the first $l$ decoder blocks, $d_l(\cdot)$, using a *local contrastive loss*, $L_l$. For a given image $x$, this loss incentivizes the representations $d_l(x) \in \mathbb{R}^{W_1 \times W_2 \times C}$ to be such that different local regions within $d_l(x)$ are dissimilar, while each local region in $d_l(x)$ remains similar across intensity transformations of $x$. Here, $W_1, W_2$ are in-plane dimensions of the feature map, and $C$ is the number of channels.

$L_l$ is defined similarly as $L_g$, but using appropriate sets of similar and dissimilar local regions from $d_l(x)$, here denoted as $\Omega^+$ and $\Omega^-$ to distinguish from the sets in the definition of $L_g$. For computing $L_l$, we feed a pair of similar images $(\tilde{x}, \hat{x})$ through the encoder ($e$), the first $l$ decoder blocks ($d_l$), and a shallow network ($g_2$) to obtain the feature maps $\tilde{f} = g_2(d_l(e(\tilde{x})))$ and $\hat{f} = g_2(d_l(e(\hat{x})))$ as illustrated in Fig. 1.ii. We divide each feature map into $A$ local regions, each of dimensions $K \times K \times C$, with $K < min(W_1, W_2)$. Now, corresponding local regions in $\tilde{f}$ and $\hat{f}$ form the similar pair set $\Omega^+$ and for each similar pair, the dissimilar set $\Omega^-$ consists of all other local regions in both feature maps, $\tilde{f}$ and $\hat{f}$. These pairs are illustrated in different colors in Fig. 1.ii. The local contrastive loss for a given similar pair is defined as:

$$l(\tilde{x}, \hat{x}, u, v) = -\log \frac{e^{\text{sim}(\tilde{f}(u,v), \hat{f}(u,v))/\tau}}{e^{\text{sim}(\tilde{f}(u,v), \hat{f}(u,v))/\tau} + \sum_{(u',v') \in \Omega^-} e^{\text{sim}(\tilde{f}(u,v), \hat{f}(u',v'))/\tau}}, \quad (3)$$

where $\text{sim}(.,.)$ is the cosine similarity as defined before, $(u, v)$ indexes local regions in the feature maps and $f(u, v) \in \mathbb{R}^{K \times K \times C}$. The total local contrastive loss for a set of images $\mathbf{X}$ can be defined as:

$$L_l = \frac{1}{|\mathbf{X}|} \sum_{x \in \mathbf{X}} \frac{1}{2A} \sum_{(u,v) \in \Omega^+} [l(\tilde{x}, \hat{x}, u, v) + l(\hat{x}, \tilde{x}, u, v)], \; \tilde{x} = \tilde{t}(x), \; \hat{x} = \hat{t}(x), \; \tilde{t}, \hat{t} \sim \mathcal{T} \quad (4)$$

where $\mathcal{T}$ is the set of transformations (intensity transformations) used to compute the local contrastive loss. For notational simplicity, we define the similar pair set using only the region indices in the summation. $L_l$, as defined above, extends $L_g$ [12] for pixel-level prediction tasks. Additionally, we may introduce domain-specific knowledge into $L_l$, as we did for the global loss in Sec. 3.1. With domain knowledge and without, we propose two strategies for constructing the sets $\Omega^+$ and $\Omega^-$.

**Random Sampling** ($L^R$): This is a direct application of the local contrastive loss without integrating domain-specific knowledge. A mini-batch is formed by randomly sampling N 2D images and applying a random pair of intensity transformations on each image. Using the notation including volume and partition indices, the random sampling strategy chooses $x_s^i$ randomly and uses $(\tilde{f}_s^i(u, v), \hat{f}_s^i(u, v))$ pairs for multiple $s$, $i$ and $(u, v)$ indices to form $\Omega^+$. For each similar pair, the dissimilar set $\Omega^-$

consists of representations of all other local regions' indices $(u', v')$ with $u' \neq u, v' \neq v$ within the same feature maps $\tilde{f}_s^i$ and $\hat{f}_s^i$.

**Strategy $L^D$**: While computing $L_g$ with strategy $L^D$, we assumed rough alignment between different volumes, defined corresponding partitions accordingly and encouraged similar global representations for such partitions. A similar strategy can be used for computing $L_l$ by assuming correspondence between local regions within images. To this end, we include additional similar pairs in $\Omega^+$ by taking corresponding local regions from different volumes, i.e. $(f_s^i(u,v), f_s^j(u,v))$, as well as their transformed versions, i.e. $(\tilde{f}_s^i(u,v), \hat{f}_s^j(u,v))$. In the dissimilar set $\Omega^-$, for each related local region pair $(u, v)$, we consider the remaining local regions $f_s^i(u', v')$ and $f_s^j(u', v')$, $u' \neq u, v' \neq v$, as well as their transformed versions. Note that the local representations extracted by $d_l$ for both the proposed strategies ($L^R, L^D$) are conceptually different than those extracted by an encoder as in [27]. Here, the local representations are designed to be distinctive across local regions.

### 3.4 Pre-training using global and local contrastive losses

We carry out the overall pre-training with the global and local contrastive losses as follows. First, we pre-train the encoder $e$ along with a shallow dense network $g_1$ using a global contrastive loss $L_g$. Next, we discard $g_1$ and use $e$ for further processing, following observations in [12]. Now, we freeze $e$ and add $l$ blocks of the decoder, $d_l$, and a shallow $g_2$ network on top of that. We then pre-train $d_l$ and $g_2$ using the local contrastive loss $L_l$. As before, we discard $g_2$ after pre-training with the local loss. After both these stages, the pre-trained $e$ and $d_l$ are expected to extract representations that capture useful information at both the global as well as local level. Now, we add the remaining decoder blocks (so that the output of the network has the same dimensions as the input) with random weights and fine-tune the entire network for the downstream segmentation task using a small labeled dataset. Note that it is possible to train with $L_g$, $L_l$, and the segmentation loss jointly, but we choose stage-wise training to avoid potentially cumbersome hyper-parameters tuning to weight each loss.

## 4 Experiments and Results

**Datasets**: For the evaluation of the proposed approach, we use three publicly available MRI datasets. **[I] The ACDC dataset** was hosted in MICCAI 2017 ACDC challenge [7, 1]. It comprises of 100 3D short-axis cardiac cine-MRIs, captured using 1.5T and 3T scanners with expert annotations for three structures: left ventricle, myocardium and right ventricle. **[II] Prostate dataset** was hosted in MICCAI 2018 medical segmentation decathlon challenge [2]. It consists of 48 3D T2-weighted MRIs of the prostate region with expert annotations for two structures: peripheral zone and central gland. **[III] The MMWHS dataset** was hosted in STACOM and MICCAI 2017 challenge [69, 70, 68, 67, 3]. It consists of 20 3D cardiac MRIs with expert annotations for seven structures: left ventricle, left atrium, right ventricle, right atrium, myocardium, ascending aorta, and pulmonary artery.

**Pre-processing**: We apply the following pre-processing steps: (i) intensity normalization of each 3D volume, $x$, using min-max normalization: $x\text{-}x_1/x_{99}\text{-}x_1$, where $x_p$ denotes the $p^{th}$ intensity percentile in $x$, and (ii) re-sampling of all 2D images and corresponding labels to a fixed pixel size $r_f$ using bi-linear and nearest-neighbour interpolation, respectively, followed by cropping or padding images with zeros to a fixed image size of $d_f$. The fixed resolutions $r_f$ and dimensions $d_f$ for each dataset are: (a) ACDC: $r_f$=1.367 $\times$ 1.367$mm^2$ and $d_f$=192 $\times$ 192, (b) Prostate: $r_f$=0.6 $\times$ 0.6$mm^2$ and $d_f$=192 $\times$ 192, (c) MMWHS: $r_f$=1.5 $\times$ 1.5$mm^2$ and $d_f$=160 $\times$ 160. We did not have to use an external tool to align volumes in any of the datasets, they were already roughly aligned as they were acquired.

Details of network architectures and optimization are provided in the Appendix.

**Experimental setup**: We split each dataset into a pre-training set $X_{pre}$ and a test set $X_{ts}$, each consisting of volumetric images and corresponding segmentation labels. We pre-train a UNet architecture using only the images from $X_{pre}$ without their labels, then fine-tune the pre-trained network with a small number of labeled examples chosen from $X_{pre}$. Fine-tuned model's segmentation performance is used to assess the pre-training procedure. The test set is not used in any stage of pre-training nor fine-tuning, it is used only in the final evaluation. The sizes of these subsets for the different datasets are: (a) $|X_{pre}|$=52, $|X_{ts}|$=20 for ACDC, (b) $|X_{pre}|$=22, $|X_{ts}|$=15 for Prostate, and (c) $|X_{pre}|$=10, $|X_{ts}|$=10 for MMWHS. For the fine-tuning stage, we form a training set $X_{tr}$ and a validation set

| Initialization of | | ACDC | | | Prostate | | | MMWHS | | |
|---|---|---|---|---|---|---|---|---|---|---|
| Encoder | Decoder | $|X_{tr}|=1$ | $|X_{tr}|=2$ | $|X_{tr}|=8$ | $|X_{tr}|=1$ | $|X_{tr}|=2$ | $|X_{tr}|=8$ | $|X_{tr}|=1$ | $|X_{tr}|=2$ | $|X_{tr}|=8$ |
| random | random | 0.614 | 0.702 | 0.844 | 0.489 | 0.550 | 0.636 | 0.451 | 0.637 | 0.787 |
| Global contrasting strategies | | | | | | | | | | |
| $G^R$ | random | 0.631 | 0.729 | 0.847 | 0.521 | 0.580 | 0.654 | 0.500 | 0.659 | 0.785 |
| $G^{D-}$ | random | 0.683 | 0.774 | 0.864 | 0.553 | 0.616 | 0.681 | 0.529 | 0.684 | 0.796 |
| $G^D$ | random | 0.691 | 0.784 | 0.870 | 0.579 | 0.600 | 0.677 | 0.553 | 0.686 | 0.793 |
| Local contrasting strategies | | | | | | | | | | |
| $G^R$ | random | 0.631 | 0.729 | 0.847 | 0.521 | 0.580 | 0.654 | 0.500 | 0.659 | 0.785 |
| $G^R$ | $L^R$ | 0.668 | 0.760 | 0.850 | 0.557 | 0.601 | 0.663 | 0.528 | 0.687 | 0.791 |
| $G^R$ | $L^D$ | 0.638 | 0.740 | 0.855 | 0.542 | 0.605 | 0.672 | 0.520 | 0.664 | 0.779 |
| Proposed method | | | | | | | | | | |
| $G^D$ | $L^R$ | **0.725** | **0.789** | **0.872** | **0.579** | **0.619** | **0.684** | **0.569** | **0.694** | **0.794** |

Table 1: Comparison of contrasting strategies (CSs) for global and local losses with number of decoder blocks set as 3 for local loss. (1) For the global loss, both $G^{D-}$ and $G^D$ are better than $G^R$ [12]. (2) For the local loss, both $L^R$ and $L^D$ are better than random decoder initialization. (3) We combine the best CSs for each loss in the proposed pre-training. Within the global and local loss results, underlines indicate the best performing CS, while the best results in each column are in bold.

$X_{vl}$, both of which are subsets of $X_{pre}$. For fine-tuning, we experiment with different sizes of the training set $|X_{tr}| = 1, 2, 8$ volumes, whereas $|X_{vl}|$ is fixed to 2 volumes.

**Evaluation**: Dice similarity coefficient (DSC) is used to evaluate the segmentation performance. For all fine-tuning experiments, we report mean scores of all structures on $X_{ts}$ over 6 runs. For each run, $X_{tr}$ and $X_{vl}$ were constructed by randomly sampling the required number of volumes from $X_{pre}$.

**Summary of experiments**: We conducted four sets of experiments, investigating: (a) the benefits of the proposed contrasting strategies in the global contrastive loss, (b) the benefits of the local contrastive loss and the two contrasting strategies, (c) the performance of the overall pre-training method (global + local contrastive losses + contrasting strategies) as compared with other pre-training, data augmentation and semi-supervised learning methods, and (d) whether the proposed pre-training method also improve performance of other techniques compared to random initialization.

**I. Contrasting strategies for global contrastive loss:** To begin with, we investigated different contrasting strategies ($G^{D-}, G^D$) for pre-training the encoder using the global contrastive loss (Eqn.1, Sec.3.1.1). After pre-training, we fine-tuned the pre-trained encoder along with a randomly initialized decoder for the segmentation task using a small number of annotated volumes ($|X_{tr}|$). The two baselines for this set of experiments are (1) no pre-training and (2) pre-training the encoder with a random contrasting strategy ($G^R$) (as in [12]). The results of this set of experiments are shown in the top part of Table 1. Firstly, we note that $G^R$ [12, 25], can be directly applied to medical images to achieve performance gains as compared to random initialization. Secondly, exploiting domain-specific knowledge of naturally occurring structure within the data with strategies $G^{D-}, G^D$ provides substantial further improvements, across all datasets and $|X_{tr}|$. These additional gains show that leveraging slice correspondence across volumes allows the network to model more complex similarity cues as compared to random augmentations, as in $G^R$. Finally, the performance with contrasting strategies is similar, but $G^D$ is slightly better than $G^{D-}$ for 6 out of the 9 settings. This indicates the benefit of leveraging domain-specific structure in the data for defining both similar and dissimilar sets. We used $G^D$ as the contrasting strategy for the global loss in further experiments.

**II. Contrasting strategies for local contrastive loss:** Next, we investigated the effect of pre-training the decoder with the local contrastive loss, $L_l$, with two contrasting strategies: $L^R, L^D$ (see Sec. 3.2.1). In order to study this independently of the global contrasting strategies, we fixed the encoder, pre-trained with $L_g$ using the random strategy $G^R$. The results of this set of experiments are shown in lower part of Table 1. We experimented with different values for the number of decoder blocks to be pre-trained, $l$, (results in appendix) and the results shown here correspond to $l = 3$, which lead to the best overall performance. It can be seen that pre-training the decoder with $L_l$ provides an additional performance boost as compared to only pre-training the encoder and randomly initializing the decoder. Importantly, such pre-training with the random strategy $L^R$ can also be used in applications when there is no obvious domain-specific clustering in the data. Further, among $L^R$ and $L^D$, it can be

| | ACDC | | | Prostate | | | MMWHS | | |
|---|---|---|---|---|---|---|---|---|---|
| Method | $\|X_{tr}\|$=1 | $\|X_{tr}\|$=2 | $\|X_{tr}\|$=8 | $\|X_{tr}\|$=1 | $\|X_{tr}\|$=2 | $\|X_{tr}\|$=8 | $\|X_{tr}\|$=1 | $\|X_{tr}\|$=2 | $\|X_{tr}\|$=8 |
| Baseline | | | | | | | | | |
| Random init. | 0.614 | 0.702 | 0.844 | 0.489 | 0.550 | 0.636 | 0.451 | 0.637 | 0.787 |
| Contrastive loss pre-training | | | | | | | | | |
| Global loss $G^R$ [12] | 0.631 | 0.729 | 0.847 | 0.521 | 0.580 | 0.654 | 0.500 | 0.659 | 0.785 |
| **Proposed init.** $(G^D + L^R)$ | 0.725 | 0.789 | 0.872 | 0.579 | 0.619 | 0.684 | 0.569 | 0.694 | 0.794 |
| Pretext task pre-training | | | | | | | | | |
| Rotation [21] | 0.599 | 0.699 | 0.849 | 0.502 | 0.558 | 0.650 | 0.433 | 0.637 | 0.785 |
| Inpainting [44] | 0.612 | 0.697 | 0.837 | 0.490 | 0.551 | 0.647 | 0.441 | 0.653 | 0.770 |
| Context Restoration [11] | 0.625 | 0.714 | 0.851 | 0.552 | 0.570 | 0.651 | 0.482 | 0.654 | 0.783 |
| Semi-supervised Methods | | | | | | | | | |
| Self-train [6] | 0.690 | 0.749 | 0.860 | 0.551 | 0.598 | 0.680 | 0.563 | 0.691 | 0.801 |
| Mixup [59] | 0.695 | 0.785 | 0.863 | 0.543 | 0.593 | 0.661 | 0.561 | 0.690 | 0.796 |
| Data Aug. [9] | 0.731 | 0.786 | 0.865 | 0.585 | 0.597 | 0.667 | 0.529 | 0.661 | 0.785 |
| Adversarial training [63] | 0.536 | 0.654 | 0.791 | 0.487 | 0.544 | 0.586 | 0.482 | 0.655 | 0.779 |
| Combination of Methods | | | | | | | | | |
| Data Aug. [9] + Mixup [59] | 0.747 | - | - | 0.577 | - | - | - | - | - |
| **Proposed init.** + Self-train [6] | 0.745 | 0.802 | 0.881 | **0.607** | **0.634** | **0.698** | **0.647** | **0.727** | **0.806** |
| **Proposed init.** + Mixup [59] | **0.757** | **0.826** | **0.886** | 0.588 | 0.626 | 0.684 | 0.617 | 0.710 | 0.794 |
| Benchmark | | | | | | | | | |
| Training with large $\|X_{tr}\|$ | | $(\|X_{tr}\| = 78)$ 0.912 | | | $(\|X_{tr}\| = 20)$ 0.697 | | | $(\|X_{tr}\| = 8)$ 0.787 | |

Table 2: Comparison of the proposed method with other pre-training, data augmentation and semi-supervised learning methods. The proposed pre-training provides better results than other methods for all datasets and $|X_{tr}|$ values, with [9] also providing similarly good results. Further, pre-training can be combined with other methods to obtain additional gains. In each column, best values among individual methods are underlined and best values overall are in bold.

seen that $L^R$ fares better for 6 out of the 9 settings across datasets and $|X_{tr}|$. Hence, we used $L^R$ for further experiments and inferred that rough alignment that is easy to obtain across volumes may not necessarily provide correspondence between local regions of different volumes, hence encouraging similarity between assumed-to-be corresponding regions may adversely affect the learning.

**III. Comparison with other methods**: Next, we compared the proposed pre-training strategy with several relevant methods, using the same architecture across all methods. As a **baseline**, we trained a network (randomly initialized) with extensive data augmentation: rotation, cropping, flipping, scaling [14], elastic deformations [51], random contrast and brightness changes [29, 45]. We found that these augmentations yield a strong baseline and used them in the fine-tuning stage for all subsequent experiments. Next, we compared with **pretext task-based pre-training** with three self-supervised tasks: rotation [21], inpainting [44] and context restoration [11], and with **global contrastive loss based pre-training** [12] where, $e$ was pre-trained according to Eq.1 (Sec.3.1.1) with a random contrasting strategy ($G^R$), while $d$ was randomly initialized. Further, we compared with data augmentation [9, 59] and semi-supervised [6, 63] learning methods that have shown promising results for medical image segmentation. Finally, we checked if the proposed initialization strategy can be combined with the other approaches of learning with limited annotations.

The results of the comparative study are shown in Table 2. Let us first consider the individual methods, i.e. without combining a pre-training strategy with a semi-supervised or data augmentation method. Among these, the proposed method that uses both global contrasting strategy with domain knowledge and local contrastive loss performs substantially better than competing methods across all datasets and $|X_{tr}|$. The performance improvements are especially high when very small number of training volumes ($|X_{tr}| = 1, 2$) are used. Comparisons with pretext-task based pre-training approaches indicate that the initialization learned with the pretext tasks provide lesser gains when fine-tuned to segmentation task in a semi-supervised setting, in comparison to proposed contrastive learning based pre-training. The results of the data augmentation [9] and adversarial training [63] methods are taken from [9] for the 2 matched datasets and the only matched setting of $|X_{tr}|$=1. We note that the data augmentation strategy suggested in [9] provides similar results as the proposed method. In the last row of Table 2, we show the fully supervised benchmark experiment, where for each dataset,

a network is trained with the maximum available training volumes. For all datasets, the proposed method reaches within ~0.1 DSC from the benchmark, with just 2 training volumes.

Finally, we observed that applying data augmentation and semi-supervised methods after the proposed pre-training rather than starting with random initialization leads to further performance gains (Table 2). Proposed pre-training combined with existing complementary augmentation and semi-supervised methods take a substantial step towards closing performance gap with the benchmark.

## 5   Conclusion

The requirement of large numbers of annotated images for obtaining good results using deep learning methods remains a persistent challenge in medical image analysis. In this work, in order to alleviate the need for large annotated training sets, we proposed several extensions in contrastive loss based pre-training [12]. Specifically, we proposed (1) a local extension of contrastive loss for learning local representations that are useful for dense prediction tasks such as image segmentation, and (2) propose problem-specific contrasting strategy that leverages naturally occurring clusters within the data to dictate similar and dissimilar image pairs that are used in the contrastive loss computation. Extensive experimentation showed that both the proposed improvements lead to substantial performance gains in the limited annotation settings for medical image segmentation in three MRI datasets. Further, we showed that the benefits conferred by the proposed initialization are orthogonal to those obtained by other methods such as data augmentation and semi-supervised learning. Overall, we believe that the proposed initialization, combined with a data augmentation technique such as Mixup [59] provides a simple toolbox for vastly improving performance in dense prediction tasks in medical imaging, especially in the clinically relevant low annotations setting.

## Acknowledgements

The presented work was partly funding by: 1. Clinical Research Priority Program Grant on Artificial Intelligence in Oncological Imaging Network, University of Zurich, 2. Swiss Platform for Advanced Scientific Computing (PASC), coordinated by Swiss National Super-computing Centre (CSCS), 3. Personalized Health and Related Technologies (PHRT), project number 222, ETH domain, 4. University Hospital Zurich. We also thank Nvidia for their GPU donation.

# 6   Broader Impact

It has been reported in the many countries [1] [2] [3] [4] that there is a scarcity of radiologists in the hospitals compared to the number of patients being imaged, thereby leading to excessive workload and consequent delays in diagnosis, prognosis and interventions. Automation of time-consuming medical imaging analyses, such as image segmentation, can assist in reducing the workload for radiologists. Supervised deep learning provides state-of-the-art performance in image segmentation on several medical imaging datasets, if large labeled datasets are available. Obtaining a large set of labeled examples from medical experts is time-consuming and expensive. In most clinical scenarios, it is not practical to expect the understaffed radiologists to dedicate time to create such large annotated sets for each new application. This expectation to create large annotated sets is a bottleneck for the deployment of current deep learning algorithms in many clinical settings. Therefore, it is crucial to develop less data-hungry algorithms that can yield high performance with few annotations.

Self-supervised learning is a promising approach to address this issue where the network is pre-trained to obtain a good initialization using the unlabeled data. Later, it is fine-tuned for a downstream task with limited annotations to yield high performance. In this work, we address this important issue by leveraging a self-supervised pre-training strategy to learn a good initialization with cheaply available unlabeled data. In this pre-training, we aim to learn useful global level and local level representations particularly useful for segmentation tasks by incorporating (1) domain-specific cues for contrasting strategies and (2) problem-specific cues with local contrastive loss to learn useful local features. We demonstrate on three medical datasets for the segmentation task that the proposed pre-training with unlabeled images can yield a good initialization, and reduce the need for large annotated sets to yield high performance when fine-tuned to the segmentation task. For instance, we get accuracy within 8% of benchmark performance by using just two 3D volumes for all datasets (for ACDC, this is just 4% of total labeled volumes) with our proposed pre-training strategy along with a simple augmentation.

Extensive validation of automated algorithms is essential before they can be used in critical decision making avenues such as healthcare. In particular, deep learning based solutions are often vulnerable to the domain shift problem, which may occur when image acquisition settings or imaging modalities are varied. Further, uncertainty quantification and interpretability may additionally be required in such systems before they can be used in practice. Nonetheless, strategies to make such systems less data hungry, such as the one proposed in this paper, are likely to constitute an important part of such systems.

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
