[Supplementary Material]

# Contrastive learning of global and local features for medical image segmentation with limited annotations

**Krishna Chaitanya**    **Ertunc Erdil**    **Neerav Karani**    **Ender Konukoglu**

Computer Vision Lab, ETH Zurich
Sternwartstrasse 7, Zurich 8092, Switzerland

`krishna.chaitanya@vision.ee.ethz.ch`

## 1  Network architecture and training details

### 1.1  Network Architecture

We use a UNet [7] based encoder ($e$) - decoder ($d$) architecture. $e$ consists of 6 convolutional blocks, each consisting of two $3 \times 3$ convolutions followed by a $2 \times 2$ maxpooling layer with stride 2. While training with global contrastive loss, we append a small network, $g_1$, on top of $e$. $g_1$ consists of two dense layers with output dimensions 3200 and 128. While training with local contrastive loss, we add a partial decoder $d_l$, with $l$ convolutional blocks, on top of the pre-trained $e$. Each block consists of one upsampling layer with a factor of 2, followed by concatenation from corresponding level of $e$ via a skip connection, followed by two $3 \times 3$ convolutions. Similar to $g_1$, we have an additional small network, $g_2$, on top of last decoder block with two $1 \times 1$ convolutions to obtain the feature map that is used for the local loss computation. Lastly, we take the pre-trained $e$ and $d_l$ and append the remaining convolutional blocks of $d$ along with skip connections such that we have a sufficient number of layers $d$ to output a segmentation with same dimensions as the input image. We use batch normalization [4] and ReLU activations after all layers except the last convolutional layers of $g_2$ and $d$.

For the computation of the local loss, the number of local regions ($A$) chosen from each feature map was 13.

### 1.2  Training Details

For each training stage, we used the Adam optimizer [5] for 10,000 iterations, with a batch size of 40 and learning rate of $10^{-3}$. For pre-training with both contrastive losses, we experimented with two values of the temperature parameter, $\tau$: 0.1 and 0.5, that provided the best performance in [1]. We observed a higher Dice score on the validation set for $\tau = 0.1$. Hence, we set $\tau = 0.1$ in all experiments. A validation set was used for model selection during the fine-tuning stage: that is, we chose the fine-tuned model that provided the highest Dice score on a validation set. Please refer to the code on github for more implementation details: https://github.com/krishnabits001/domain_specific_cl.

### 1.3  Dataset and other training details

*(a) Data pre-processing*: All images are bias-corrected using N4 [8] bias correction ITK toolkit.

*(b) Data split*: The data split was chosen so as to have the number of volumes for pre-training ($X_{pre}$) and testing ($X_{ts}$) to be roughly 50% of each dataset. For clarity, we present the data split numbers in table 1. For Prostate, although we have 48 volumes, labels were provided only for a subset of them, so the number of volumes for each set were adjusted accordingly. For ACDC, we updated the benchmark training with $|X_{tr}| = 78$ instead of 50 where we obtained test DSC of 0.912 that is similar to 0.908, obtained with earlier evaluated configuration of $|X_{tr}| = 50$.

| Dataset | Total no. of volumes | $\|X_{pre}\|$ | $\|X_{ts}\|$ | $\|X_{tr}\|$ for benchmark |
|---------|------------|------------|------------|------------|
| ACDC | 100 | 52 | 20 | 78 (+2 val) |
| Prostate | 48 | 22 | 15 | 20 (+2 val) |
| MMWHS | 20 | 10 | 10 | 8 (+2 val) |

Table 1: Dataset split

Figure 1: A 2D slice is taken from each of the four partitions, across three different volumes, with the partition number indicated by $s$. Here, each row presents four images from four partitions of a selected volume.

*(c) Validation set $X_{vl}$:* We use $X_{vl}$ fixed to 2 3D volumes during fine-tuning to determine when to stop the training, used for final evaluation on the test set.

*(d) Fine-tuning:* As mentioned in main article, we experiment with 3 settings: $|X_{tr}|$ = 1, 2, and 8 3D volumes with $X_{vl}$ fixed to 2 3D volumes.

## 1.4 Details of training time and convergence

On a Titan X GPU, the training time is approximately: (a) 2 hours for $L_g$ pre-training, (b) 4 hours for $L_l$ pre-training, and (c) 2 hours for fine-tuning. Also, we found the pre-training convergence to be consistently stable.

## 2 Illustration of slice correspondence in medical volumetric images

We illustrate 2D slices taken from each of the four partitions, from three different volumes in Fig. 1, where the partition number is indicated by $s$. In this figure, even though we see changes in shape and intensity characteristics for 2D slices of the same partition from different volumes, they contain the same global information about the cardiac anatomy.

# 3 Ablation studies

Here, we present an ablation study to investigate the effect of some of the hyper-parameters involved in the proposed method. For all these experiments, we considered the proposed global loss contrasting strategy $G^D$ for pre-training the encoder as it yielded the best performance in earlier experiments. We used the ACDC dataset for these experiments.

## 3.1 Global Contrastive Loss

Firstly, we investigated the effect of batch size and the number of partitions per volume $S$ used in the pre-training of the encoder with global contrastive loss.

### 3.1.1 Batch Size

Here, we studied the effect of the batch size used during the encoder pre-training. Previous works have suggested that large batch sizes are crucial for good performance, with some works leveraging memory bank [9] or momentum contrast [3, 6] to accommodate higher number of negative samples information per batch. In order to check if the same is applicable for our datasets as well, we pre-trained the encoder with 3 batch sizes: 40, 250, 450, with the number of partitions $S$ set to 4.

The results are presented in Table 2. We observed that for medical images, higher batch sizes do not improve the results any further, rather the performance deteriorated for the batch size of 450. Unlike natural image datasets, for the evaluated medical imaging datasets we observe that we may not require large batch sizes in the pre-training stage to obtain high performance. Further evaluation on more datasets is required to arrive at a more conclusive statement on the effect of batch size on medical datasets.

Moreover, medical datasets generally contain a much lower number of unlabeled images compared to natural image datasets that can be leveraged for pre-training. For instance, the largest of the 3 datasets used in our experiments is the ACDC dataset with 100 volumes amounting to around 1000 2D images. So, evaluating batch size values like 2048, 4096 used in natural image datasets for pre-training may not be a practical setting for medical images.

| Batch Size | $|X_{tr}|$=1 | $|X_{tr}|$=2 | $|X_{tr}|$=8 |
|---|---|---|---|
| 40 | **0.691** | **0.784** | **0.870** |
| 250 | 0.685 | 0.779 | 0.862 |
| 450 | 0.668 | 0.770 | 0.857 |

Table 2: Mean dice score over test set ($X_{ts}$) shown on ACDC dataset for the effect of batch size with the number of partitions set ($S$) to 4 in the pre-training of the encoder with $G^D$ strategy and later is fine-tuned to all training set sizes ($|X_{tr}| = 1, 2, 8$).

### 3.1.2 Number of partitions per 3D volume

Here, we evaluated 3 values for number of partitions ($S$) per 3D volume $S=3, 4, 6$ for a fixed batch size of 40. The number of partitions determines the number of clusters that are formed in the latent space for the proposed global loss strategy $G^D$. The results are presented in Table 3. We observed that the dice score degraded for higher values of $S$. As there are approximately 10 images per volume in ACDC data, using higher values of $S$ will force the network to create a cluster for each image in the volume. This results in the unrealistic composition of both positive and negative pairs of images across volumes, where an inaccurate association of dissimilar images will be forced to act as similar pairs due to a higher value of $S$. Since the volumes are not perfectly aligned, this wrong association in representation space leads to a subsequent drop in performance.

## 3.2 Local Contrastive Loss

Secondly, we investigated the effect of decoder size ($d_l$), local region size ($K \times K$) that are used in the proposed local contrastive loss for pre-training the decoder network. We considered the proposed local contrastive loss strategy $L^D$ to pre-train the decoder blocks with the encoder kept frozen. The encoder was earlier pre-trained with global loss strategy $G^D$.

| No. of partitions ($S$) per 3D volume | $|X_{tr}|$=1 | $|X_{tr}|$=2 | $|X_{tr}|$=8 |
|---|---|---|---|
| 3 | 0.686 | 0.776 | 0.868 |
| 4 | **0.691** | **0.789** | **0.870** |
| 6 | 0.652 | 0.760 | 0.859 |

Table 3: Mean dice score over test set ($X_{ts}$) shown on ACDC dataset for the effect of the number of partitions $S$ per 3D volume for a fixed batch size of 40 in the pre-training of the encoder with $G^D$ strategy and later is fine-tuned to all training set sizes ($|X_{tr}| = 1, 2, 8$).

### 3.2.1 Decoder size

We varied the number ($l$) of decoder blocks that are pre-trained. We investigated all five values: $l = \{1, 2, 3, 4, 5\}$, where $l = 5$ ($d_5$) means that the entire decoder is pre-trained. From Table 4, we observe the performance for the number of decoder blocks of $l = 1, 2, 3$ yielded higher gains compared to $l = 4, 5$. We hypothesize that this is because, at $l = 5$, local regions have a smaller receptive field compared to other $l$ values and thereby might not contain enough information to learn useful representations.

| Region size | $|X_{tr}|$=1 | | | | | $|X_{tr}|$=2 | | | | |
|---|---|---|---|---|---|---|---|---|---|---|
| $K \times K$ | $d_1$ | $d_2$ | $d_3$ | $d_4$ | $d_5$ | $d_1$ | $d_2$ | $d_3$ | $d_4$ | $d_5$ |
| $1 \times 1$ | 0.738 | 0.704 | 0.705 | 0.702 | 0.683 | 0.780 | 0.777 | 0.770 | 0.774 | 0.764 |
| $3 \times 3$ | 0.703 | 0.737 | 0.725 | 0.726 | 0.690 | 0.783 | 0.793 | 0.787 | 0.776 | 0.777 |

Table 4: Mean dice score over test set ($X_{ts}$) on ACDC data for the proposed contrasting strategies of $G^D$ and $L^D$ used to pre-train the encoder and a different number of decoder blocks ($d_l, l = 1, 2, 3, 4, 5$) for two different values of local region sizes ($K \times K$) of $1 \times 1$ and $3 \times 3$ and later is fine-tuned to two training set sizes ($|X_{tr}| = 1, 2$).

### 3.2.2 Size of local regions

We experimented with 2 values: $K = 1 \times 1$ and $K = 3 \times 3$, for the size of local region used to obtain local representations in a given feature map. This was to study if the size of the local regions influences the performance post pre-training. From Table 4, we observe a small difference in the performance of around 2% between the two region sizes considered for $d_3$, with $3 \times 3$ size yielding higher performance. This can be because $3 \times 3$ local region contains more information due to a higher receptive field that is potentially more useful in the devised pre-training setting.

Additionally, we ran this ablation experiment on the remaining datasets (with $d_l = 3$, and sampling strategies $G^D$, $L^D$). Table 5 presents these results which show that local region size $3 \times 3$ works better for most settings, as seen with ACDC.

### 3.3 Combination of Local and Global Contrastive Losses

Here, we present the combinations of local contrastive loss strategies ($L^R$, $L^D$) with both global loss strategies ($G^R$, $G^D$) for different decoder block $d_l$ lengths (l=2,3,4) that is moved from the Table 1 in the main article to Table 6.

| Dataset | $K \times K$ | $|X_{tr}|$=1 | $|X_{tr}|$=2 |
|---|---|---|---|
| Prostate | $1 \times 1$ | 0.554 | 0.614 |
| | $3 \times 3$ | 0.567 | 0.607 |
| MMWHS | $1 \times 1$ | 0.559 | 0.674 |
| | $3 \times 3$ | 0.574 | 0.681 |

Table 5: Effect of local region sizes ($K \times K$) on downstream segmentation performance in DSC.

| Initialization of Encoder | Decoder | Dataset | $\lvert X_{tr}\rvert=1$ | | | $\lvert X_{tr}\rvert=2$ | | | $\lvert X_{tr}\rvert=8$ | | |
|---|---|---|---|---|---|---|---|---|---|---|---|
| | | | $d_l=2$ | $d_l=3$ | $d_l=4$ | $d_l=2$ | $d_l=3$ | $d_l=4$ | $d_l=2$ | $d_l=3$ | $d_l=4$ |
| Local loss strategies $L^R, L^D$ on encoder pre-trained with random strategy $G^R$ | | | | | | | | | | | |
| $G^R$ | random init | ACDC | 0.631 | | | 0.729 | | | 0.847 | | |
| $G^R$ | $L^R$ | | 0.642 | **0.668** | 0.655 | 0.754 | **0.760** | 0.732 | **0.860** | 0.850 | **0.860** |
| $G^R$ | $L^D$ | | 0.614 | 0.638 | 0.642 | 0.744 | 0.740 | 0.744 | 0.854 | 0.855 | 0.852 |
| $G^R$ | random init | Prostate | 0.521 | | | 0.580 | | | 0.654 | | |
| $G^R$ | $L^R$ | | **0.566** | 0.557 | 0.538 | 0.600 | 0.601 | 0.591 | 0.661 | 0.663 | 0.665 |
| $G^R$ | $L^D$ | | 0.536 | 0.542 | 0.543 | 0.583 | **0.605** | 0.597 | 0.656 | **0.672** | 0.659 |
| $G^R$ | random init | MMWHS | 0.500 | | | 0.659 | | | 0.785 | | |
| $G^R$ | $L^R$ | | 0.523 | **0.528** | 0.511 | 0.692 | 0.687 | 0.679 | 0.794 | 0.791 | 0.792 |
| $G^R$ | $L^D$ | | 0.510 | 0.520 | 0.515 | **0.697** | 0.664 | 0.684 | **0.797** | 0.779 | 0.781 |
| Local loss strategies $L^R, L^D$ on encoder pre-trained with proposed strategy $G^D$ | | | | | | | | | | | |
| $G^D$ | random init | ACDC | 0.691 | | | 0.784 | | | 0.870 | | |
| $G^D$ | $L^R$ | | 0.708 | 0.725 | 0.720 | 0.784 | 0.789 | 0.785 | 0.868 | 0.872 | 0.871 |
| $G^D$ | $L^D$ | | **0.737** | 0.725 | 0.726 | **0.793** | 0.787 | 0.776 | 0.865 | **0.874** | 0.868 |
| $G^D$ | random init | PZ | 0.579 | | | 0.600 | | | 0.677 | | |
| $G^D$ | $L^R$ | | 0.577 | 0.579 | **0.581** | 0.617 | 0.619 | **0.620** | 0.683 | 0.684 | 0.685 |
| $G^D$ | $L^D$ | | 0.562 | 0.567 | 0.564 | 0.608 | 0.607 | 0.599 | 0.675 | **0.686** | 0.680 |
| $G^D$ | random init | MMWHS | 0.553 | | | 0.686 | | | 0.793 | | |
| $G^D$ | $L^R$ | | 0.556 | 0.569 | 0.572 | 0.671 | **0.694** | 0.693 | 0.796 | 0.794 | 0.796 |
| $G^D$ | $L^D$ | | 0.545 | **0.574** | 0.551 | 0.677 | 0.681 | 0.689 | **0.803** | 0.791 | 0.789 |

Table 6: Mean Dice score over $X_{ts}$ for the proposed local contrastive loss on all datasets for the decoder lengths $d_l, l = 2, 3, 4$ and all training set sizes $\lvert X_{tr}\rvert=1,2,8$.

| $\lambda_l$ | $\lvert X_{tr}\rvert=1$ | $\lvert X_{tr}\rvert=2$ |
|---|---|---|
| 1 | 0.634 | 0.741 |
| 10 | 0.633 | 0.730 |
| 100 | 0.643 | 0.745 |
| 1000 | 0.644 | 0.739 |

Table 7: Results on ACDC dataset when using joint pre-training to train both encoder and decoder in 1 step instead of 2 steps.

### 3.4 Stage-wise training vs joint training

Results with joint training are shown in Table 7. We define the total loss: $L_{net} = L_g + \lambda_l * L_l$, where $\lambda_l$ is a hyper-parameter to balance loss values. As per R4's idea, the encoder weights are updated with the net loss $L_{net}$ that includes $L_l$, unlike our stage-wise training, where only $L_g$ was used to update the encoder. We tried 4 values of $\lambda_l$ on ACDC dataset for $d_l=3$. Results indicate that stage-wise training (where DSC is 0.725 for $\lvert X_{tr}\rvert=1$ and 0.789 for $\lvert X_{tr}\rvert=2$) performs better.

## 4 Experiments with Natural Image Datasets

In order to check the generality of the proposed method beyond medical imaging datasets, we evaluated the proposed local contrastive loss on a natural image dataset "Cityscapes" [2] for the segmentation task. We pre-trained the decoder using the proposed version of local contrastive loss ($L^R$) and the encoder with global contrastive loss (random strategy $G^R$) as in [1]. We compared it to a baseline with no pre-training and pre-training with only global contrastive loss as in [1]. Additionally, we also evaluated the combination of proposed initialization along with Mixup [10] that yielded the best results on medical images.

For the evaluation, we split the whole training and validation data provided into $X_{pre}$ (2770 images) and $X_{ts}$ (705 images) like earlier. We used this test set $X_{ts}$ only for the final evaluation. After pre-

training with only images of $X_{pre}$ (no labels are used), we fine-tuned the network for segmentation task with a set of labeled images $X_{tr}$ and validation images $X_{val}$ chosen from $X_{pre}$ ( $X_{tr}, X_{val} \subset X_{pre}$). We performed the fine-tuning in a limited annotation setting for three values of $X_{tr}, X_{val} = \{(100,100), (200,200), (400,200)\}$. Table 8 presents these results.

To implement the global and local contrastive losses we need a reasonable batch size value of around 40. Due to memory issues, it was difficult to implement such a batch size with images in their original dimensions (1024,2048). Therefore, we down-scaled the images by a factor of 4 to (256,512). Due to the downsampling, some of the smaller objects either vanished or were reduced to a negligible size. In order to avoid extreme class imbalance issues, we set these small objects as background. Thus, we considered the following 12 foreground labels: road, sidewalk, building, wall, vegetation, terrain, sky, person+rider, car, motorbike+bicycle, truck, bus, with the remaining labels set as background.

**Training details**: For augmentation, we used random cropping followed by random color jitter (brightness, contrast, saturation, hue). Rest of the training details remain same as described for medical imaging datasets.

| Method | $X_{tr}, X_{val} =$(100,100) | $X_{tr}, X_{val} =$(200,200) | $X_{tr}, X_{val} =$(400,200) |
|---|---|---|---|
| Baseline (Random init.) | 0.451 | 0.495 | 0.524 |
| Global loss $G^R$ [1] | 0.457 | 0.496 | 0.535 |
| **Proposed init.** $(G^R + L^R)$ | 0.469 | 0.517 | 0.549 |
| **Proposed init.** + Mixup [10] | 0.475 | 0.526 | 0.569 |
| Benchmark | 0.652 | | |

Table 8: Mean dice score over test set ($X_{ts}$) for all the selected labels on Cityscapes [2] dataset for the proposed pre-training compared to a baseline with the random initialization, and pre-training with only global loss.