[Reviews · NeurIPS 2020]

Review 1

Summary and Contributions: Authors propose a contrastive learning approach for semi-supervised segmentation of 3D volumes. Particularly, to this end, authors integrate medical imaging context in a global contrastive loss and propose a local contrastive loss on the features of the decoder to improve local representations. Evaluation is done on several public segmentation benchmarks and compared to several other self-training and semi-supervised methods.

Strengths: - Results are competitive. - The paper is easy to read. - The idea of local contrastive loss for segmentation is interesting.

Weaknesses: - Despite the idea is somehow interesting, this paper just brings a new strategy to generate additional positive and negative pairs for contrastive loss. - Furthermore, this methodology is tailored to the situation of aligned pairs of images. Which is the performance of all the compared methods when images are not aligned? Or how sensitive the performance is to bad registrations? - I feel additional ablation studies are required. For example, in the local contrastive loss, the regions are sampled based on K (< min (W,H)), but I assume the value of K will be dependent on the application, as the context needed to capture the target across different datasets. However, authors show results only on ACDC. - Some of these datasets contain multiple classes. Nevertheless, whether the DSC is computed over one or multiple targets (and then averaged) is not explained. If multiple classes are used, how the sampling for local contrastive loss is performed? Does the fact the the same patch may contain information from several targets affect the local structure representation? - In some point of the paper, authors claim that the only use 2 images for training. As such, I think authors should also report the performance of the method in [63], as they achieve satisfactory results with only 1 training image. - According to results in Table 1, it seems that the local contrastive term without the domain specific knowledge (L_R) typically achieves better performance than the term integrating domain knowledge (L_D). - As a minor comment, it is hard to get the whole message when Table 1 in the main paper and Table 4 in supplemental material are supposed to show the same thing.

Correctness: The claims and methodology are correct. According to the results, I would revise, however, the contribution related to the domain-specific knowledge, particularly in the local contrastive term.

Clarity: The paper is clearly written.

Relation to Prior Work: Yes, prior work is properly discussed and authors state the differences with respect to those works.

Reproducibility: Yes

Additional Feedback: ***** Update after the rebuttal phase ****** After reading carefully the authors' rebuttal, the other reviewers and the paper again, I am more positive about this work. Even though I feel the technical contribution is somehow limited, authors propose a smart way to leverage structured data in tasks involving dense prediction, i.e., segmentation. This strategy obtains satisfactory results and can serve as an interesting baseline to use contrastive learning in semantic segmentation on medical images.


Review 2

Summary and Contributions: This paper proposes a contrastive learning model for medical image segmentation leveraging both domain-specific and problem-specific cues. For extracting the domain-specific cue, two novel strategies are proposed to leverage the spatial information through-slice direction. For extracting the problem-specific cue, a local version contrastive loss is introduced to leverage the in-slice spatial information.

Strengths: a. The whole paper is written clearly and logically b. Sufficient experiments to prove the effectiveness of the proposed ideas c. The sketch figure is drawn pretty well and really helps the reader understand the idea. d. The method also stands out in terms of novelty. The proposed contrastive learning method wells capture the properties of medical images: 1) Proposing strategies for leveraging both in-slice and through-slice spatial information. 2) Introducing problem-specific loss in contrastive learning to improve performance.

Weaknesses: a. In Line 145-148, I doubt whether the expression of "images" is accurate. From my perspective, "images" should be replaced with "partitions". b. Some experiment settings need more explanations. 1) The motivation for setting the validation set should be explained. Is it set for tuning any hyperparameters? 2) The datasets are stated to have 100,20 and 48 cases, however, 50,20 and 8 cases are used as the benchmark experiment (in Table 2) setting. An explanation and selection details should be given. 3) The reason to have different training and testing ratios for different datasets should be given. c. The setting of local region size (K x K) is recommended to be given in the main part rather than the supplementary part.

Correctness: claims and method correct; the empirical methodology correct

Clarity: Yes

Relation to Prior Work: Yes

Reproducibility: Yes

Additional Feedback: update: The authors have well addressed my concerns in the rebuttal. I will keep my initial score


Review 3

Summary and Contributions: The paper addresses the problem of learning to segment medical images with limited data. It does so using pre-training using contrastive learning with three contributions: a new domain-specific contrasting strategy for volumetric images, a local version of contrastive loss, encouraging local region representations to be similar under different transformations, and evaluations of these contributions in comparison to other techniques for learning with limited annotations. The contributions are evaluated on three tasks with data and benchmark results publicly available and in comparison with other state of the art techniques for learning with limited data (pretext task self-supervision, data-augmentation etcetera). Results are consistently better with the proposed techniques.

Strengths: The problem of segmenting medical images with limited annotations to learn from is highly relevant. The proposed techniques are relatively simple, yet good ideas that I have not seen previously. Claims appear to be sound (with a small exception in the supplement) and based on a relatively thorough empirical evaluation. ** Update after author rebuttal ** - The author reviews and the author rebuttal has not changed my initial positive look on the paper, the small exception has been addressed (C2). Table 1 and 2, would be good to have in the supplement or in the paper.

Weaknesses: There are a number of small weakness to the approach, the technique to some degree depends on well registered images and there are a number of extra hyper parameters introduced, such as the number of partitions to use per 3D volume, the number of pre-trained decoder blocks to use and the region size. I would expect these aspects to be largely problem dependent, and the degree to which results would also be improved on other problems is therefore somewhat unclear. ** Update after author rebuttal ** - The authors clarified that no image registration was performed, which indicates that alignment is either good enough on these datasets or that the approach is not very sensitive to misalignment. I do not think this invalidates the above comment. Aside from the approach itself, I would also have liked some information on training time and convergence. How easy is this to setup, train and add to existing training processes? ** Update after author rebuttal ** - Some useful details on this has been provided (B3) and the authors will make the code public which will help with reproducibility. The pre-training is done stage-wise (section 3.4), where the encoder is first trained with the proposed global contrastive loss, and then afterwards the decoder is trained with the proposed local contrastive loss with encoder weights fixed. This choice seems relatively odd and the only reason given is to avoid extra hyper parameters that would otherwise be needed to weight global and local contrastive losses. One could imagine, however, that the obtained weights for the encoder become less ideal for generating relevant local features at the decoding stage. It would be good to know if the authors experimented with joint training of the global and local losses. ** Update after author rebuttal ** - A table showing the results of joint training has now been provided (Table 2). Interestingly this appears to do much worse than stage-wise training. The table is helpful, would be good to have in the supplement, and validates the use of the stage-wise training the authors used originally. Being an annoying reviewer, one could ask for motivation for why the authors chose these specific values of lambda. One could also suggest using stage-wise followed by joint training, to help the network find a better starting point for the joint training. Never-the-less, I appreciate the work the authors did to satisfy my criticism. The approach deals with volumetric images, but takes two-dimensional images as input and while segmenting medical volumes by processing two-dimensional image slices individually can often give quite good results, some 3D processing would likely improve results. Could the contrastive loss be teaching the network something about the 3D structure, which makes it better at dealing with the limited input? If so then the proposed techniques might not lead to similar improvements if the network made predictions based on 3D information. ** Update after author rebuttal ** - The authors seem to agree with this assessment (A7). This is perhaps something to explore in future work.

Correctness: Both methods and claims in the paper seem reasonable. I have not found any relevant errors in the empirical methodology. I find the statement (in the supplement) "This shows that unlike natural image datasets, medical images do not require large batch sizes in the pre-training stage to obtain high performance." too strong given empirical evidence of just three datasets. ** Update after author rebuttal ** - This point has been addressed (C2).

Clarity: I found some parts of the paper to be hard to read (see detailed comments), mostly due to the manuscript being relatively densely written. The few mistakes and unclear aspects that are there thus made understanding a lot harder.

Relation to Prior Work: Yes contributions are clearly mentioned and relevant prior work pointed out as far as I can see.

Reproducibility: Yes

Additional Feedback: - In lines 104-105 it is written "The presence of g_1(ยท) allows e() some flexibility to also retain information regarding the transformations, as was empirically shown in [12]". Could you elaborate on what you mean by this and why this is helpful? - You use D to indicate the number of images (line 136) as well as to denote strategy, e.g. G^{D}, maybe consider a different symbol. - The strategy of splitting volumes into a number of partitions seems a bit artificial. As I understand it, the last image in a partition would be assumed to be dissimilar to the next image in the following partition. Despite the closeness of such two images. To me it would make more sense to define similarity of images based on how close they are in the volume. Is there a particular reason for this choice? - The textual description of the strategies 'G^{D-}' and 'G^{D}' are a bit unclear. E.g. writing 'Here, restrict contrasting images from corresponding partitions in different volumes as such images...' does not really describe what you are doing well. - The abbreviations used for the different strategies G^D, G^{D-}, etcetera could be explained. E.g. why use a D? This would make them easier to remember. - Line 176, what is W_1, W_2 and C? (Again in line 183, with K) - In equation (3) and surrounding text you do not define the role of \tilde{x} and \hat{x}. Also the \tilde{f} and \hat{f} now take two arguments compared to their definitions at line 181. While I still believe I understand the details, it complicates understanding. - Detail regarding the network architecture, is given in the supplementary material. It would be good to mention this in the paper. - Not including corresponding partitions and corresponding local regions from different volumes in the contrastive loss is probably a good idea and relatively safe, whereas, assuming corresponding partitions and corresponding local regions from different volumes to be similar is likely less safe. The latter is likely more sensitive to alignment errors and biological differences between subjects, particularly when it comes to the local regions. The authors specifically evaluate the dissimilar (D-) and combined dissimilar and similar strategy (D) for the global contrastive loss but not the local contrastive loss. It would have been interesting to see the same done for the local contrastive loss, where the authors only evaluate the combined strategy (D). I would assume the local contrastive loss to be more sensitive to alignment errors and biological differences, which could explain why L^D mostly performs worse than L^R. Line 303, "that" -> "than" Supplement, Table 3, "we observe an increase in the performance upon increasing the number of decoder blocks until the value of l = 3, and then it goes down again.", there are 4 experiments with 5 decoder blocks, none of them has the maximal value at l = 3 as far as I can see. ** Update after author rebuttal ** - The authors do not address each of these points in detail, but I am happy they appreciate them as a whole. There are a couple of these points that I would have liked more feedback to such as the splitting strategy and the local loss being evaluated with a dissimilar and similar strategy as well. But these are more questions to satisfy my curiousity than criticism of the work.

[Author Response · NeurIPS 2020]

We thank the reviewers for their time and detailed comments. All reviewers appreciated the novelty of the method and
the thoroughness in the experiments. We categorize the concerns of the reviewers and the corresponding responses into
the following 3 groups: A. regarding the method, B. regarding experiments and evaluation, and C. miscellaneous.

*A1. Contributions [R1]*: The main concern of R1 is that the paper *just* proposed a new strategy for selecting +ve and -ve
pairs for the contrastive loss. We emphasize that, as acknowledged by R3, R4, the paper has 3 main contributions: we
(1) leverage domain knowledge to form appropriate +ve and -ve pairs, leading to clear gains over random augmentations
as done in prior works [12], (2) propose a local contrastive loss useful for dense prediction tasks like segmentation and
(3) show that pre-training is complementary to semi-supervised and data augmentation methods.

*A2. Dependence on registration [R1, R4]*: The method requires only rough alignment across volumes. This can be
obtained with very basic registration, even using the transformation matrices located in the header files of medical
images without an external registration algorithm. As a demonstration, in all the experiments presented in the article,
we **did not** perform any registration and used volumetric images as they were distributed in the challenge datasets.

*A3. Effect of multiple classes within a local region [R1]*: $L_l$ does not take any label information into account. So, even
when a local region consists of several labels, its representation contains information about the entire local region. $L_l$
seeks to make this representation consistent across various intensity transformations and simultaneously be different
from other distant local regions within the image.

*A4. Effect of domain-specific knowledge in local loss $L_l$ [R1, R4]*: $L_l$ is a novel loss proposed by us, which improves
performance as compared to only using the global loss $L_g$ (as seen in Table 1, row 5 in the main article). We further
propose and study the effect of two sampling strategies within $L_l$: (a) $L^D$, where local regions are matched across
volumes (referred to as using domain knowledge), and (b) $L^R$, which does not assume such correspondences. Our
experiments show that $L^R$ performs better than $L^D$. We believe that this is not a drawback of the method, but instead,
an indication that obtaining perfect in-plane alignment across volumes is difficult due to inter-subject variability (also
pointed out by R4). We view $L^R$ as a contribution of the proposed work.

*A5. Effect of local region size ($K \times K$) [R1]*: We ran this ablation experiment on the remaining datasets (with $d_l = 3$,
and sampling strategies $G^D, L^D$). Results (Table 1) show that $3 \times 3$ works better for most settings, as seen with ACDC.

*A6. Stage-wise v/s joint training [R4]*: Results with joint training are shown
in Table 2. We define the total loss: $L_{net} = L_g + \lambda_l * L_l$, where $\lambda_l$ is a hyper-
parameter to balance loss values. As per R4's idea, the encoder weights
are updated with the net loss $L_{net}$ that includes $L_l$, unlike our stage-wise
training, where only $L_g$ was used to update the encoder. We tried 4 values
of $\lambda_l$ on ACDC dataset for $d_l$=3. Results indicate that stage-wise training
(where DSC is 0.725 for $|X_{tr}|$=1 and 0.789 for $|X_{tr}|$=2) performs better.

| Dataset | $K \times K$ | $|X_{tr}|$=1 | $|X_{tr}|$=2 |
|---|---|---|---|
| Prostate | $1 \times 1$ | 0.554 | 0.614 |
| | $3 \times 3$ | 0.567 | 0.607 |
| MMWHS | $1 \times 1$ | 0.559 | 0.674 |
| | $3 \times 3$ | 0.574 | 0.681 |

Table 1: (A5) Effect of local region size.

*A7. Relevance of the method for 3D CNNs [R4]*: We agree that the proposed pre-training
($L_g, G^D$) may be informing the 2D CNN about the 3D structure of medical images. We
believe that this is beneficial as compared to training 3D CNNs, where one faces memory
issues as well as has more risk of overfitting due to a higher number of parameters.

| $\lambda_l$ | $|X_{tr}|$=1 | $|X_{tr}|$=2 |
|---|---|---|
| 1 | 0.634 | 0.741 |
| 10 | 0.633 | 0.730 |
| 100 | 0.643 | 0.745 |
| 1000 | 0.644 | 0.739 |

Table 2: (A6) Joint training.

*B1. Experimental setup*: *(a) Data split [R3]*: The data split was chosen with the idea of
keeping the number of volumes for pre-training ($X_{pre}$) and testing ($X_{ts}$) to be roughly
around 50% of each dataset. For Prostate, although we have 48 volumes, labels were
provided only for a subset of them, so the number of volumes for each set were adjusted accordingly. For ACDC,
we ran the benchmark training with $|X_{tr}| = 78$ instead of 50 and obtained test DSC of 0.912, comparable to 0.908
obtained with $|X_{tr}| = 50$. We are happy to add these details in the revised supplementary. *(b) Validation set $X_{vl}$ [R3]:*
We use $X_{vl}$ fixed to 2 3D volumes during fine-tuning to determine when to stop the training.*(c) Fine-tuning [R1]:* As
mentioned in line 251, we experiment with 3 settings: $|X_{tr}| = 1, 2$, and 8 3D volumes with $X_{vl}$ fixed to 2 3D volumes.

*B2. Comparison with [63] [R1]*: We compare with [9], also based on data augmentation (like [63]), but more general in
that it does not depend on a deformable registration step, which is difficult to achieve for anatomies other than the brain.

*B3. Details of training time and convergence [R4]*: On a Titan X GPU, training takes about: (a) 2 hours for $L_g$
pre-training, (b) 4 hours for $L_l$ pre-training, and (c) 2 hours for fine-tuning. Also, we found the pre-training convergence
to be consistently stable. We will add these details in the revised supplementary and also make the code public.

*C1. Clarity in notation [R4]:* We really appreciate the detailed comments provided by R4. We will incorporate the
suggested notational changes and required additional details in the revised version.

*C2. Writing [R4]:* We agree that the comment regarding batch size for pre-training with medical images is too strong.
We will tone this down appropriately in the revised version.

[Meta-Review · NeurIPS 2020]

Three knowledgeable referees agree that the paper makes a simple yet valuable contribution, which holds the potential of being broadly used in the medical imaging domain. They recognize the soundness of the proposed approach and its compelling experimental validation. I agree with the referees and recommend acceptance.